# Dysregulation of miR-223, miR-146a, and miR-193a Expression Profile in Acute and Chronic Phases of Experimental Autoimmune Encephalomyelitis in C57BL/6 Mice

**DOI:** 10.3390/cells13171499

**Published:** 2024-09-06

**Authors:** Saba Gharibi, Bahram Moghimi, Mohammad Bagher Mahmoudi, Ensieh Shahvazian, Ehsan Farashahi Yazd, Maryam Yadegari, Mohammad Taher Tahoori, Esmaeil Yazdanpanah, Dariush Haghmorad, Valentyn Oksenych

**Affiliations:** 1Institute for Physical Activity and Nutrition, School of Exercise and Nutrition Sciences, Deakin University 1 Gheringhap Street, Geelong, VIC 3220, Australia; 2Department of Genetics, Faculty of Medicine, Shahid Sadoughi University of Medical Sciences, Yazd, Iran; 3Research and Development Department, ROJETechnologies, Yazd, Iran; 4Department of Biology and Anatomical Sciences, Faculty of Medicine, Shahid Sadoughi University of Medical Sciences, Yazd, Iran; 5Department of Immunology, Faculty of Medicine, Shahid Sadoughi University of Medical Sciences, Yazd, Iran; 6Department of Immunology, School of Medicine, Semnan University of Medical Sciences, Semnan, Iran; 7Broegelmann Research Laboratory, Department of Clinical Science, University of Bergen, 5020 Bergen, Norway

**Keywords:** multiple sclerosis, acute phase, chronic phase, EAE, MiR-223, MiR-146a, MiR-193a

## Abstract

Multiple sclerosis (MS) is a chronic autoimmune disease with an unknown etiology. The purpose of this research was to assess miR-223, miR-146a, and miR-193a in acute and chronic phases of experimental autoimmune encephalomyelitis (EAE) mice to consider the possible role of these genes in the pathogenesis of MS. EAE induction was given by myelin oligodendrocyte glycoprotein peptide on female C57BL/6 mice. Clinical scores and other criteria were followed daily until day 21 for the acute group and day 77 for the chronic group. At the end of the course, inflammation and demyelination of the central nervous system (CNS) were assessed by histological analysis. MicroRNA expression levels were assessed by real-time PCR. EAE development attenuated in the chronic group, and histological analysis showed less infiltration and demyelination in the chronic group compared to the acute group. The upper expression of miR-223 is demonstrated in the acute phase of EAE. Moreover, the expression levels of miR-146a and miR-193a decreased in the chronic phase of EAE. MiR-223 showed a highly coordinated elevation in the acute phase both in vivo and in vitro. MiR-146a shares a pathway with miR-223 through effecting IL-6 expression. Further studies are needed to reveal their impact on EAE and possible applications as drug targets and biomarkers.

## 1. Introduction

Multiple sclerosis (MS), a debilitating autoimmune disorder in young people, characterized by chronic inflammation, demyelination, and neurodegeneration within the central nervous system (CNS) [1]. Experimental autoimmune encephalomyelitis (EAE) is a widely used animal model for studying the pathogenesis of MS [2]. EAE is induced by immunization of animals with myelin antigens or adoptive transfer of encephalitogenic T cells, leading to an immune-mediated attack on CNS myelin [3]. The disease course in EAE is marked by distinct phases, including an acute inflammatory phase characterized by rapid onset of neurological deficits and extensive immune cell infiltration, followed by a chronic phase where ongoing neuroinflammation contributes to progressive neuronal damage and disability [4]. Although various theories have been made for the pathogenesis of EAE, it is commonly accepted that EAE is the result of a cascade of inflammatory events triggered by aberrant autoreactive Th1 and Th17 and then activated microglia and resident macrophages in the CNS [5]. The regulation of immune responses in EAE and MS is influenced by various molecular mechanisms, including the expression of microRNAs (miRNAs), which are small non-coding RNAs that play critical roles in gene regulation [6]. Among these, miR-223, miR-146a, and miR-193a have been implicated in modulating immune responses and inflammation [7,8].

miR-223 is highly expressed in myeloid cells, including macrophages, dendritic cells, and neutrophils, which are integral components of the innate immune system. This miRNA is involved in the differentiation and activation of these cells, influencing their ability to produce pro-inflammatory cytokines, phagocytose pathogens, and present antigens to T cells [9]. In the context of EAE, miR-223 has been shown to modulate the inflammatory response by regulating the expression of key genes involved in immune cell activation and migration [10].

miR-146a serves as a crucial negative feedback regulator within the immune system, particularly in controlling the activation of the NF-κB signaling pathway. NF-κB plays a central role in the induction of inflammatory cytokines, chemokines, and adhesion molecules in response to various stimuli, including microbial products and cytokines [11]. By targeting key components of the NF-κB pathway, such as IL-1 receptor-associated kinase 1 (IRAK1) and tumor necrosis factor receptor-associated factor 6 (TRAF6), miR-146a helps to dampen excessive inflammation, thereby preventing tissue damage [12]. In autoimmune diseases like MS and EAE, reduced expression or functional impairment of miR-146a can result in uncontrolled inflammation, leading to increased disease severity and chronicity [13].

miR-193a is another miRNA with important roles in immune regulation and cellular homeostasis. It is involved in the regulation of apoptosis as well as cell proliferation [14]. In the CNS, balanced apoptosis is essential for eliminating autoreactive immune cells and preventing excessive tissue damage during autoimmune responses. miR-193a has been shown to influence the expression of genes involved in apoptotic pathways, thereby contributing to the regulation of cell survival during inflammation [15]. Dysregulation of miR-193a may lead to the persistence of autoreactive immune cells or insufficient repair of damaged tissues, exacerbating the neurodegenerative aspects of EAE.

The specific expression patterns and regulatory functions of miR-223, miR-146a, and miR-193a in the different phases of EAE suggest that they play distinct and potentially complementary roles in the disease process [16]. Understanding how these miRNAs are dysregulated during the acute and chronic phases of EAE could provide valuable insights into the molecular mechanisms underlying the transition from acute inflammation to chronic neurodegeneration. This knowledge could also highlight new therapeutic targets for modulating miRNA expression to alleviate symptoms or slow the progression of autoimmune diseases like MS.

## 2. Materials and Methods

### 2.1. Animals

Female C57BL/6 mice (eight weeks old) were purchased from the Pasteur Institute of Iran (Tehran, Iran). Mice were located in animal facilities with normal conditions, including a temperature of 25 °C, relative humidity of 50%, and a 12 h light/dark period. All mice had free access to standard rodent chow and filtered water. All animal experiments were performed according to Shahid Sadoughi University of Medical Sciences’ ethical guidelines (approval code: IR.SSU.REC.1378.121, approval date: 15 June 2020). The experiment was planned in three groups: healthy normal group (*n* = 8); acute group (*n* = 7); and chronic group (*n* = 8).

### 2.2. Induction of EAE

EAE was induced by immunizing C57BL/6 mice subcutaneously in the flank with 300 µg myelin oligodendrocyte glycoprotein (MOG_35–55_) (Sigma-Aldrich, St. Louis, MO, USA) emulsified in complete Freund’s adjuvant (Sigma-Aldrich, St. Louis, MO, USA) consisting of 5 mg/mL M. tuberculosis H37RA (Difco Laboratories, Detroit, MI, USA). At the time of immunization, and 48 h later, mice were injected intravenously with 250 ng pertussis toxin (Sigma-Aldrich, St. Louis, MO, USA) [17,18]. Clinical manifestations of EAE were followed, and the weight of the mice was calculated daily until 25 days post-immunization. Mice were scored for neurologic impairment under the following scale: 0, no clinical symptom; 1, limited failure of tail tonicity; 2, complete failure of tail tonicity; 3, flabby tail and unusual form of walking; 4, hind limb paralysis; 5, hind limb palsy with hind body partial paralysis; 6, hind and foreleg paralysis; 7, moribund or death [19,20]. All EAE-induced and healthy mice were placed in similar conditions. On day 21, seven mice were randomly chosen for the acute group and prepared for further experiments; we chose day 21 for an acute phase, and 8 mice were selected for the chronic group and continued until day 77.

### 2.3. Histological Analysis

To evaluate the amount of CNS inflammation and demyelination at the date of sacrifice (at days 21 and 77 after EAE induction for acute and chronic groups, respectively), mice were anesthetized using a combination of ketamine (150 mg/kg) and xylazine (10 mg/kg). Subsequently, their brains were removed, and intracardiac perfusion with PBS containing 4% paraformaldehyde was performed. Paraffin-embedded 5 mm sections of the brain and spinal cord were stained with hematoxylin and eosin (H&E) for assessment of inflammation and Luxol fast blue (LFB) for demyelination. Myelin loss in tissue sections was visualized by a loss of LFB staining. Sections are analyzed by a light microscope (Olympus, Tokyo, Japan) in a blinded manner. Inflammation was determined as follows: 0. No inflammation; 1. A small number of inflammatory cells; 2. Existence of perivascular infiltrates; 3. Extending intensity of perivascular cuffing with spread into contiguous tissue. Demyelination was scored as follows: 0. No demyelination; 1. Unique foci; 2. Few areas of demyelination; 3. Considerable areas of demyelination [21,22].

### 2.4. Cell Culture and Nucleic Acid Preparation

Peripheral lymph nodes (inguinal and axillary), spleens, and brain were cut out from C57BL/6 mice at days 21 and 77 after EAE induction for the acute and chronic groups, respectively. Red blood cells were depleted using ammonium chloride. Cell suspensions were prepared and cultured in RPMI 1640 medium consisting of 10% fetal bovine serum (FBS), 100 U/mL penicillin, and 100 mg/mL streptomycin (all reagents obtained from Sigma, St. Louis, MO, USA) in round-bottom 24-well plates (2 × 10^6^ cells/well). The tissues of each mouse were separately pulverized in phosphate buffer using a nylon mesh to isolate cells. The suspension was centrifuged at 3000 g for 10 min, and the producing cell pellet was suspended in a Trizol™ RNA isolation kit (Thermo Fisher Scientific, Gloucester, UK) for RNA extraction. cDNA synthesis was performed by the Revert Aid First Strand cDNA Synthesis Kit (Thermo Fisher Scientific, Gloucester, UK) from total RNA using gene-specific stem-loop reverse transcription primers. Specific primers for target genes, including

MiR146-a (5′GTCGTATGCAGAGCAGGGTCCGAGGTATTCGCACTGCATACGACaaccca3′), 

MiR-193a (5′GTCGTATGCAGAGCAGGGTCCGAGGTATTCGCACTGCATACGACactggg3′), 

MiR-223 (5′GTCGTATGCAGAGCAGGGTCCGAGG TATTCGCACTGCATACGACactggg3′) and

SnoR-202 (5′GGTCGTATGCAGAGCAGGGTCCGAGGTATCCATCGCACGCATCGCACTGCATACGACCTCTCAG3′).

### 2.5. Real-Time PCR

Real-time PCR was carried out in Rotor-Gene Q (Qiagen, Hilden, Germany). Amplification was performed in a total volume of 10 μL for 40 cycles, and products were identified using SYBR Green I dye (Applied Biosystems, Thermo Fisher Scientific, Gloucester, UK). Samples were run in duplicate, and the relative expression level was measured by normalization to the endogenous gene SnoR-202, with results displayed as relative expression units. Real-time PCR primer sequences were miR146a, forward: 5′-GCAATGAGAACTGAATTCCAT-3′, reverse: 5′-GAGCAGGGTCCGAGGT′; miR-193a, forward: 5′-CAACTGGCCTACAAAGTCC-3′, reverse: 5′-GAGCAGGGTCCGAGGT-3′, miR-223, forward: 5′-GCTGTCAGTTTGTCAAATACC-3′, reverse: 5′-GAGCAGGGTCCGAGGT-3′, SnoR-202, forward: 5′-AGATTTAACAAAAATTCGTCAC-3′, reverse: 5′-GAGCAGGGTCCGAGGT-3′.

Based on an interpretation of target genes normalized to SnoR-202, we measured the relative quantification delta-delta CT, and the results are displayed as fold change related to normal healthy mice.

### 2.6. Statistical Analysis

One-way, non-parametric analysis of variance (ANOVA) (Kruskal–Wallis test) followed by Dunn’s multiple comparison tests was directed for analysis of clinical symptoms between groups. A comparison of the groups, on the progress of clinical signs, was conducted via two-way repeated-measures ANOVA. Mann–Whitney non-parametric unpaired t-tests were applied for two-group comparisons. SPSS 21 was utilized to analyze the data. Data were displayed as the mean ± SEM. Statistical difference was agreed at the level of *p* < 0.05.

## 3. Results

### 3.1. EAE Development Exacerbated the Clinical Manifestations

Institute for Physical Activity and Nutrition, School of Exercise and Nutrition Sciences, Deakin University, Geelong, Australia. In the present research, 8- to 10-week-old C57BL/6 mice were immunized with MOG_35–55_ peptide together with mycobacterium tuberculosis and pertussis toxin to induce a chronic progressive form of EAE. Fifteen mice were randomly located in acute and chronic groups, and 8 mice were placed in the healthy normal group. After immunization with MOG, the first clinical symptom observed in the mice was a failure of tonicity in the tail. The disease symptoms started earlier in the EAE-induced mice (day 9). Parameters of onset of clinical symptoms, the severity of disease, clinical outcome, and a cumulative score of disease varied in acute and chronic groups. Moreover, the mice in acute and chronic groups lose body weight compared to the healthy normal group (Figure 1 and Table 1).

### 3.2. Immune Cell Infiltration into the CNS and Demyelination of the Brain Increased during EAE

To determine inflammation in the CNS and axonal damage, we evaluated inflammatory cell infiltration and demyelination of the brain during EAE. Hematoxylin and eosin (H&E) staining was performed to visualize infiltrating immune cells. Fewer infiltrating leukocytes and sections of perivascular cuffing were detected in the chronic group. A notable increase in leukocyte infiltration in the brain was demonstrated in the acute group (Figure 2).

To evaluate demyelination, brain sections were stained by Luxol fast blue. Here, we detected significant demyelination in the areas of the brain and a reduction in myelin density during the progression of disease in the acute group. In the chronic group, results showed moderate demyelination compared to the acute group, which indicates remyelination in this group (Figure 3).

### 3.3. The Greater Expression of miR-223 Observed in the Acute Phase of EAE Compared to a Chronic Phase

To further explore the probable role of molecular mechanisms in the pathogenesis of EAE, we considered the expression of microRNAs such as miR-223, miR-193a, and miR146a in the brain, spleen, and lymph nodes. Expression data of the target microRNAs were normalized with a corresponding mean value of the internal gene, SnoR-202, formerly recognized as a proper reference gene in the same conditions.

In the spleen, miR-223 expression was significantly higher in an acute group compared to chronic and healthy normal groups (*p* < 0.001). Moreover, brain expression of miR-223 was considerably higher in an acute group compared to chronic and healthy normal groups (*p* < 0.05 and *p* < 0.01, respectively). Similar results were detected in lymph nodes of the acute group compared to chronic and healthy normal groups (Figure 4 and Table 2).

In MOG-stimulated cultured cells from spleen and lymph nodes, miR-223 expression was significantly increased in an acute group compared to chronic and healthy normal groups (*p* < 0.01) (Figure 4 and Table 2). 

### 3.4. The miR-193a Expression Pattern in the Brain Decreased in the Chronic Phase of EAE

In the spleen, a significant reduction in expression of miR-193a was observed in both chronic and acute groups compared to the healthy normal group (*p* < 0.001). In the brain, a significant increase was observed in a chronic group compared to both acute and healthy normal groups (*p* < 0.001). In lymph nodes, the miR-193a expression level was significantly enhanced in an acute group compared to both chronic and healthy normal groups (*p* < 0.001 and *p* < 0.05, respectively) (Figure 5 and Table 2).

In MOG-stimulated cells cultured from spleen and lymph node cells, the expression level of miR-193a in the acute group displayed a significant increase compared to chronic and healthy normal groups (spleen *p* < 0.001, and lymph nodes *p* < 0.05) (Figure 5 and Table 2).

### 3.5. The Expression Pattern of miR-146a Declined in the Chronic Phase of EAE

MiR-146a expression level in the spleen was significantly diminished in the chronic group matched to the acute group (*p* < 0.01). No significant difference was observed between acute or chronic groups in contrast with the healthy normal group. In addition, in the brain and lymph nodes, no significant distinction was observed in the miR-146a expression level (Figure 6 and Table 2).

To explore the possible role of miR-146a expression level in EAE acute and chronic phases, we compared the expression level of miR-146a in MOG-stimulated cultured cells from spleen and lymph nodes in single-cell suspensions, re-stimulated with MOG_35–55_ antigen in vitro. Samples were prepared from mice in all groups. 

As illustrated in Figure 6, miR-146a expression in cultured cells from the spleen showed a significant decline in the chronic phase connected to acute and healthy normal groups (*p* < 0.05). The same result was discovered in a chronic group compared to acute and healthy normal groups in lymph node cells (*p* < 0.05).

## 4. Discussion

A growing body of evidence pinpoints that dysregulation in the epigenetic machinery is correlated with the pathogenesis of MS [23]. It is expected that microRNAs have been demonstrated to be involved in the differentiation, development, synaptic maturation, and plasticity of the neurons [24]. Recently, microRNAs have become the focus of attention because alterations in the expression level of different microRNAs are broadly engaged in the pathogenesis of neurodegenerative diseases, especially MS [7,25]. These findings lead to the hypothesis that microRNAs can be applied as a reliable new biomarker in the screening, diagnosis, and therapy monitoring of various diseases [7]. The present study reported three microRNAs with altered expression in vivo and in vitro in acute and chronic phases of EAE and normal mice: miR-223, miR-146a, and miR-193a, which are regarded as key managers of the immune system in MS.

MiR-223 serves as a myeloid cell-specific microRNA that is also expressed in the nervous system; therefore, upregulated miR-223 may disrupt the protective roles against neuronal cell death in nervous system disorders [26]. Immunologically, miR-223 has been confirmed to trigger the differentiation of CD4^+^ towards Th1 and Th17 cells, the leading cell type in MS pathology, and a defect of miR-223 results in resistance to EAE in mice [27]. Additionally, miR223^−/−^ DCs have less efficiency to support Th1 and Th17 differentiation, as well as the expression of inflammatory cytokines such as IL-17 and IFN-γ, TNF-α, and IL-1β and IL-6 remarkably decreased in the spinal cords of miR-223 deficient mice [27]. On the other hand, the engagement of miR-223 in the pathogenesis of EAE was also supported by evidence from miR-223 knock-in mice, which exhibit a delayed onset of the paralytic disease and a significant reduction in EAE severity, which was accompanied by a reduction in Th1 and Th17 responses [27]. Therefore, this finding indicates that the pathogenesis of MS and EAE involves miR-223 as an enhancing DC cell activation, a regulator of T cell activation and proliferation, as well as Th1/Th17 differentiation [27].

The increased expression of miR-223 in the spleen, lymph node, brain, and MOG-induced cultured cells in the acute phase is linked to the chronic phase and healthy normal mice, implying that miR-223 might perform an essential task in the expansion and evolution of MS. In accordance with our study, Satoorian et al. have observed elevated expression of miR-223 not only in the spinal cord but also in the spleen, lymph nodes, and bone marrow of EAE-bearing mice compared with healthy normal mice [27]. In another study by using qRT-PCR analysis, Hosseini et al. showed that the levels of CD4^+^ T-cell-derived miR-223 significantly enhanced during the relapsing phase of MS compared to the remitting phase and healthy individuals; therefore, miR-223 was accompanied by increased Th17 cells and decreased Treg cells, a potential role in positively regulating the pathogenic cascade that leads to EAE [28]. Furthermore, the upregulation of miR-223 in CD4^+^ T cells of MS patients and EAE models [29], a greater degree of miR-223 in PBMCs from MS patients correlated with healthy people [30], as well as a higher rank of miR-223 in active MS lesions compared to normal CNS areas in healthy subjects, was recently published by another group in accordance with our findings [31]. Given the studies above, the difference in miR-223 expression is an essential determining cause of MS pathogenesis and is described as potential biomarkers, therapeutic agents, or drug targets in patients with MS.

MiR-146a is defined as a crucial regulator of immune response and inflammation and further has been noted to be related to the hazard of multiple autoimmune disorders [7]. It is dramatically induced in response to pro-inflammatory cytokines such as TNF-α, IL-1, and lipopolysaccharide (LPS) and subsequently targeted the NFĸB pathway through inhibition of IL-1 receptor-associated kinase 1 (IRAK-1) and TNF receptor-associated factor 6 (TRAF-6), which are essential for pro-inflammatory signaling [32]. Fourth more, it becomes proven that miR-146a is a negative regulator of the IFN pathway [33]. These findings supported the anti-inflammatory effects of miR-146a in the pathogenesis of autoimmunity disorder, even contributing to being a potential biomarker of an inflammatory condition, diagnosis, or treatment effectiveness in these diseases. In another study, low expression of miR-146a was detected in cases with systemic lupus erythematosus (SLE) related to healthy people, and a negative correlation was noted between serum miR-146a levels and disease activity [34]. This is supported by previous studies indicating that miR-146a is engaged in the suppression of the inflammatory process. However, some studies revealed that upregulated miR-146a is correlated with the peak of disease and performs a pivotal role in eliciting and maintaining the inflammation [7,33,35].

Several recent studies have explored the involvement of miR-146a in MS and its model animal. In our present study, the expression level of miR-146a was enhanced in the acute phase compared to the chronic phase in the spleen as well as MOG-stimulated cells cultured from spleen and lymph nodes, but in the brain no significant difference was observed.

In the Lescher et al. study, consistent with the present study, the miR-146a expression level was elevated in marmoset and mouse EAE lesions [36]. In the Ma et al. study, the analysis of real-time quantitative PCR revealed the expression of miR-146a significantly increased in both PBMCs and brain white matter lesions from MS patients [7]. Similar to MS and EAE, a high concentration of miR-146a expression has been reported in other autoimmune diseases, such as rheumatoid arthritis (RA), SLE, and Sjögren’s syndrome patients [37]. Notably, there is a positive relationship between the high levels of miR-146a expression in peripheral blood mononuclear cells and active disease because the low expression level of miR-146a is correlated with inactive disease in patients with RA [38]. These lines of evidence, closely with our results, support a potential role of miR-146a as an inducer of inflammatory or autoimmunity responses in the pathogenesis of MS/EAE.

To examine possible reasons for a difference in the upregulation of miR-146a and its immunologic role in autoimmunity disorders, several theories are suggested. One theory is that high levels of miR-146a expression due to unclear mechanisms do not correlate with its negative regulator effect on the IFN pathway and do not lead to feedback control of autoimmune inflammation [33]. Interestingly, the high level of miR-146a expression was strongly correlated with levels of IL-17 [39] and TNF-α expression [35]. Another possible theory is that the increased miR-146a may become less effective in precisely regulating its targets, resulting in insufficient modulation of TRAF-6 and IRAK-1. This could lead to prolonged production of pro-inflammatory cytokines and the activation of adaptive immune responses [33,37].

MiR-193a has been revealed to apply anti-tumor and anti-cancer effects and is downregulated in the majority of primary cancer tissue [40]. The expression and biological function of miR-193a in the pathology of inflammatory and autoimmune diseases remain mostly unexplored. Further studies are required to define and illuminate the role of microRNAs in MS. To our knowledge, this is the original study to explore the expression of miR-193 in EAE.

MiR-193a has been demonstrated to be upregulated in CD4^+^, CD8^+^ T cells, and B cells of peripheral blood in the remission phase of MS patients linked to healthy volunteers [41]. Additionally, upregulation of miR-193a has been observed in both PBMCs and brain white matter lesions from patients with MS as compared with the normal brain white matter tissues [7]. In our study, the expression profile of miR-193a was different between the immune organs and the brain. We observed a remarkably decreased expression level of miR-193a in the spleen of the acute phase group, while a raised level of miR-193a was discovered in the lymph nodes. These contradictory results demonstrated that MiR-193a might have a dual role in the molecular pathogenesis of EAE.

MiR-193a may have both inflammatory and anti-inflammatory effects. It seems that miR-193a is suppressed at the end of the acute phase in the spleen, while lymph nodes have not responded and miR-193a is elevated in lymph nodes. However, the brain revealed a unique expression pattern, and highly increased levels of miR-193a expression were discovered in the chronic phase, which might indicate a protective effect for miR-193a in the brain. Differences observed in previous studies and our study may be due to differences in the cell content of the studied tissues and/or differences in the pathogenesis of EAE and MS and/or differences in the targets of miR-193a in humans and mice.

## 5. Conclusions

In this study, we observed dysregulation of miR-223, miR-193a, and miR-146a in EAE. miR-223 showed a highly coordinated, significant elevation in the acute phase both in vivo and in vitro, which makes it a reliable target for further studies on molecular mechanisms of EAE and MS and further drug target and biomarker studies. miR-146a showed an elevated expression in the spleen in the acute phase and shares a pathway with miR-223, through effecting IL-6 expression, while miR-193a showed dual regulation in the spleen, lymph nodes, and brain. All studied miRNAs have significant dysregulation, which needs further studies to fully reveal their impact on EAE and their possible applications as drug targets and biomarkers.

## Figures and Tables

**Figure 1 cells-13-01499-f001:**
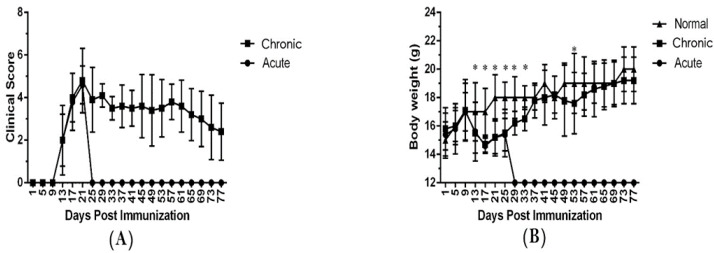
EAE development exacerbated the clinical symptoms. EAE induced in female C57BL/6 mice with MOG_35–55_ as detailed under Materials and methods. Mice were monitored for signs of EAE, and the results for all mice, were presented as (**A**) mean clinical score ± SEM, and (**B**) body weight for acute, chronic and healthy normal groups. Results were expressed as mean ± SEM. * *p* < 0.05. Mice were divided into three groups: (1) acute group; (2) chronic group; (3) healthy normal group.

**Figure 2 cells-13-01499-f002:**
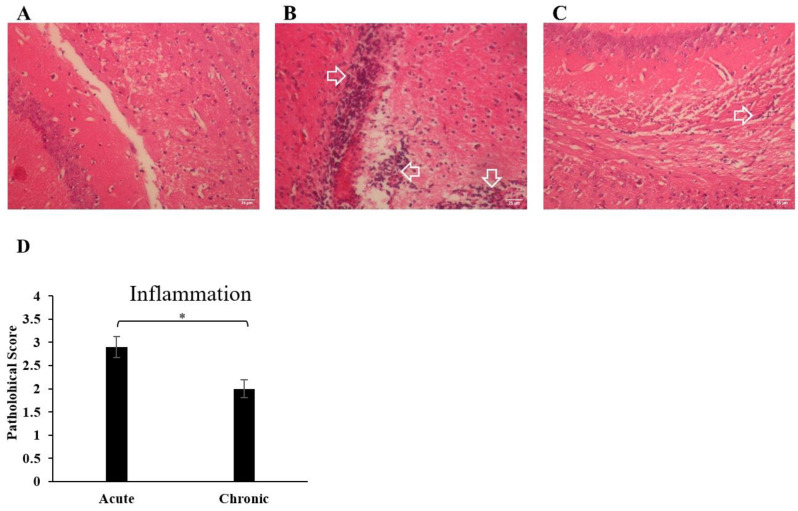
Immune cell infiltration into the CNS increased during EAE. A histopathological assessment of the brain from each group was performed. Brains from acute and healthy normal groups were collected on day 21 and chronic group on day 77 post-immunization, fixed in paraformaldehyde and embedded in paraffin. Five micrometer sections from different regions of the brain from each of the groups were stained with H&E to enumerate infiltrating leukocytes; (**A**) healthy normal group, a very low number of infiltrated leucocytes was observed. (**B**) Acute group, there was a prominent increase in infiltrated leucocytes. The infiltrated leucocytes are shown by the arrow. (**C**) Chronic group, there was a decrease in the infiltrated leucocytes. The infiltrated leucocytes are shown by the arrow. (**D**) Pathological scores of inflammation were analyzed and shown with a bar graph as mean scores of pathological inflammation ± SEM. Data represent three independent experiments. * *p* < 0.05.

**Figure 3 cells-13-01499-f003:**
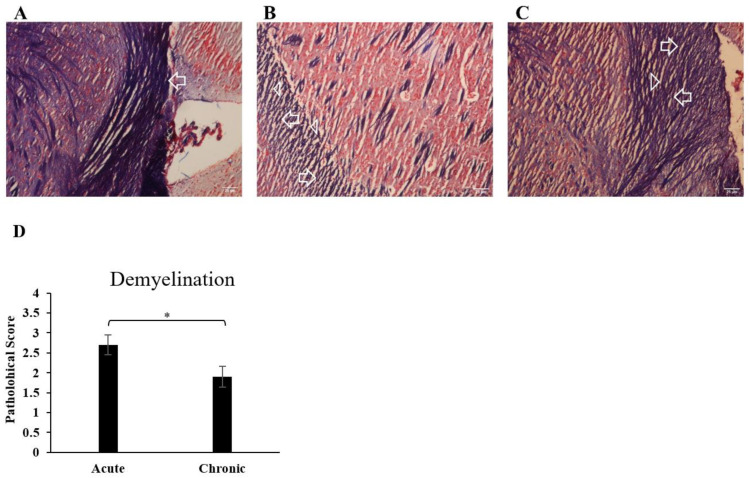
Demyelination of the brain increased during EAE. Sections from different regions of the brain from each of the groups were stained with Luxol fast blue to assess demyelination. CNS inflammatory foci and infiltrating inflammatory cells were quantified. (**A**) The healthy normal group showed normal blue staining of LFB. (**B**) Acute group, the blue color of LFB was greatly reduced, which indicates a high level of myelin loss in this group. Myelin density is shown by an arrow, and myelin loss is shown by an arrowhead. (**C**) Chronic group, the myelin loss is considerably lower than in the acute group. Myelin density is shown by an arrow, and myelin loss is shown by an arrowhead. (**D**) Pathological scores of demyelination were analyzed and shown with a bar graph as mean scores of demyelination ± SEM. Data represent three independent experiments. * *p* < 0.05.

**Figure 4 cells-13-01499-f004:**
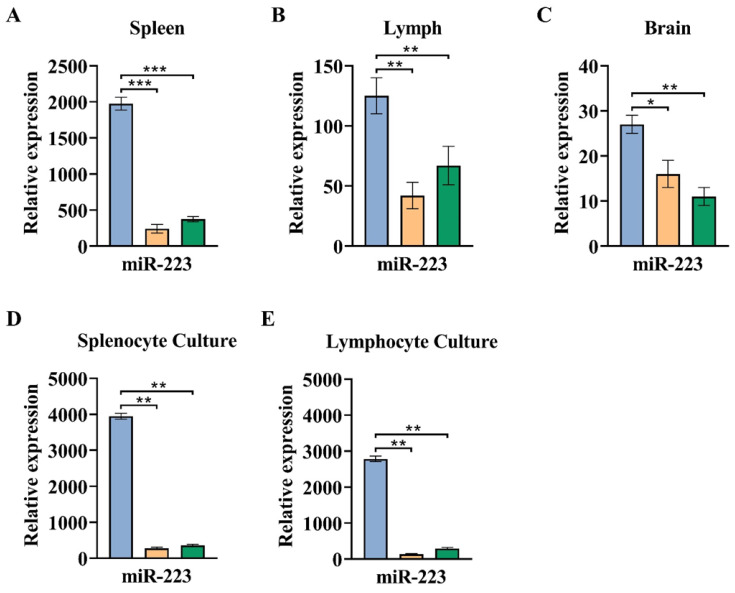
The higher expression of miR-223 was observed in the acute phase of EAE compared to the chronic phase in vivo and in vitro. On days 21 (acute group) and 77 (chronic group) post-immunization, spleen, brain, and lymph nodes were collected and for in vitro study re-stimulated with MOG35–55. MicroRNA level was assessed by real-time quantitative PCR as described in the Material and Methods section. (**A**) The expression level of miR-223 in spleen cells for acute, chronic, and healthy normal groups. (**B**) The expression level of miR-223 in lymph nodes. (**C**) The expression level of miR-223 in the brain. (**D**) The expression level of miR-223 in cells cultured from the spleen of all groups. (**E**) The expression level of miR-223 in cells cultured from lymph nodes. Results are normalized relative to the expression level of reference gene snor-202 and were expressed as mean ± SEM. The assay was run in triplicate, and the results are normalized relative to the expression level of reference gene snor-202 and were expressed as mean ± SEM. * *p* < 0.05, ** *p* < 0.01, *** *p* < 0.001 compared with normal group.

**Figure 5 cells-13-01499-f005:**
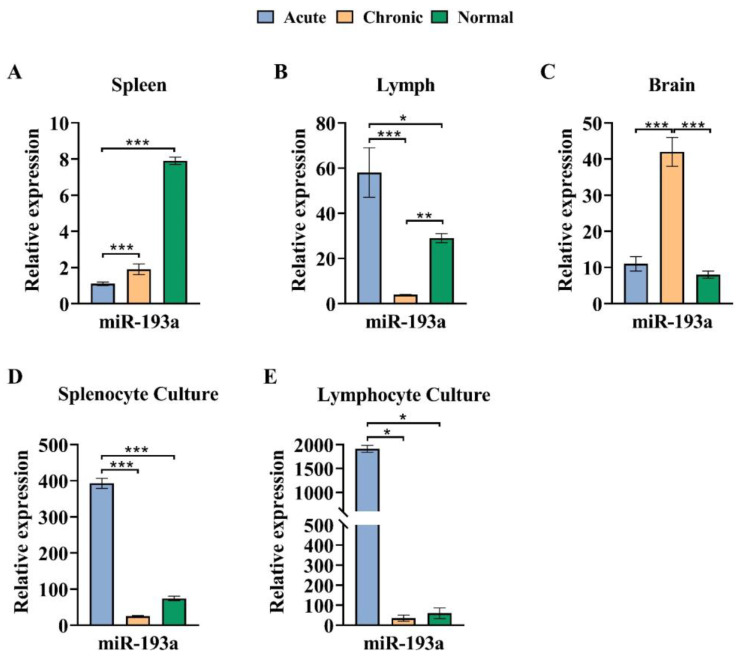
The miR-193a expression pattern in the brain increased in the chronic phase of EAE in vivo and in vitro. (**A**) Expression level of miR-193a in spleen cells for acute, chronic, and healthy normal groups. (**B**) The expression level of miR-193a in lymph nodes. (**C**) The expression level of miR-193a in the brain. (**D**) Expression level of miR-193a in cells cultured from the spleen of all groups. (**E**) The expression level of miR-193a in cells cultured from lymph nodes. The assay was run in triplicate, and the results are normalized relative to the expression level of reference gene snor-202 and were expressed as mean ± SEM. * *p* < 0.05, ** *p* < 0.01, *** *p* < 0.001 compared with normal group.

**Figure 6 cells-13-01499-f006:**
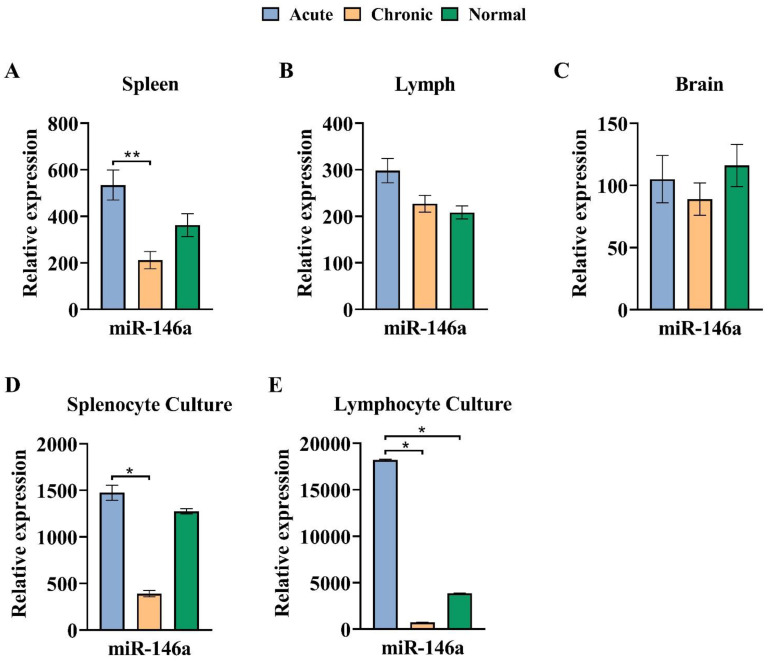
The expression pattern of miR-146a declined in the chronic phase of EAE in vivo and in vitro. (**A**) Expression level of miR-146a in spleen cells for acute, chronic, and healthy normal groups. (**B**) The expression level of miR-146a in lymph nodes. (**C**) The expression level of miR-146a in the brain. (**D**) The expression level of miR-146a in cells cultured from the spleen of all groups. (**E**) The expression level of miR-146a in cells cultured from lymph nodes. The assay was run in triplicate, and the results are normalized relative to the expression level of reference gene snor-202 and were expressed as mean ± SEM. * *p* < 0.05, ** *p* < 0.01 compared with normal group.

**Table 1 cells-13-01499-t001:** Clinical features of EAE.

Group	Maximum Score	*p*-Value	Mean Score (Last Day)	*p*-Value *
Acute	4.8 ± 0.83 (*n* = 7)	*p* < 0.05	4.8 ± 0.83 (*n* = 7)	*p* < 0.05
Chronic	4.8 ± 0.52 (*n* = 8)		2.4 ± 1.3 (*n* = 8)	

Data were expressed as mean ± SD. * *p*-value < 0.05 is considered to be statistically significant.

**Table 2 cells-13-01499-t002:** Significant results of miR-223, miR-193a, and miR-146a expression in examined circumstances.

Tissue/Cell		Spleen			Lymph Nodes			Brain			Cells Cultured from Spleen			Cells Cultured from Lymph Nodes		
miRNA	Disease phase	Acute	Chronic	Normal	Acute	Chronic	Normal	Acute	Chronic	Normal	Acute	Chronic	Normal	Acute	Chronic	Normal
miR-223	Acute		⬆	⬆		⬆	⬆		⬆	⬆		⬆	⬆		⬆	⬆
	Chronic	⬇		-----	⬇		-----	⬇		-----	⬇		-----	⬇		-----
miR-146a	Acute		⬆	-----		-----	-----		-----	-----		⬆	-----		⬆	⬆
	Chronic	⬇		-----	-----		-----	-----		-----	⬇		⬇	⬇		-----
miR-193a	Acute		-----	⬇		⬆	⬆		⬇	-----		⬆	⬆		⬆	⬆
	Chronic	-----		⬇	⬇		⬇	⬆		⬆	⬇		-----	⬇		-----

All arrows indicate a significant change of column compared to row.

## Data Availability

All data are included in this manuscript.

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
