# Peer review of "Dysregulation of miR-223, miR-146a, and miR-193a Expression Profile in Acute and Chronic Phases of Experimental Autoimmune Encephalomyelitis in C57BL/6 Mice"

_cells, 2024, doi:10.3390/cells13171499_

Round 1

Reviewer 1 Report

Comments and Suggestions for Authors

The authors presented the expression profile of genes: miR-223, miR-146a, and miR-193a  in acute and chronic phases of experimental autoimmune encephalomyelitis (EAE) mice to consider the possible role in the pathogenesis of multiple sclerosis.

Comments:

Introduction: needs complete revision. In general, there is too much introduction of MS and then the explanation of the animal model of MS. The introduction should concentrate on the 3 main genes in the title.

Line 35: it cannot be said that MS is the reason for anything. This is a disease. This should be rephrased. Lines 41-42: MS can have a wide range of symptoms, not only these few. In general, the introduction should not explain too much about MS. Maybe more EAE and definitely genes.

Methods:

Animals: there should be written Approval of the Ethical Committee (name and number as well).

How much Ketamine & Xylazine for anesthesia?

Results: photomicrographs in Figures 2 and 3 are too dark. Please, make them brighter.

Figures 4,5 and 6: the fonts are too small. Please increase them.

The conclusion needs its subheading.

Comments on the Quality of English Language

moderate

Author Response

Q1. Introduction: needs complete revision. In general, there is too much introduction of MS and then the explanation of the animal model of MS. The introduction should concentrate on the 3 main genes in the title.

Line 35: it cannot be said that MS is the reason for anything. This is a disease. This should be rephrased. Lines 41-42: MS can have a wide range of symptoms, not only these few. In general, the introduction should not explain too much about MS. Maybe more EAE and definitely genes.

A1. We acknowledge the reviewers' recommendation to update the introduction and incorporate more background on EAE and the relevant genes. In response to this feedback, we have revised the manuscript to include additional relevant information in the introduction. The updated content has been highlighted.

Methods:

Q2. Animals: there should be written Approval of the Ethical Committee (name and number as well).

A2. We thank the reviewer for this comment, the Ethical Committee with code and date of approval added to the manuscript.

Q3. How much Ketamine & Xylazine for anesthesia?

A3. We thank the reviewer for this comment, the manuscript has been revised accordingly, and the following content has been incorporated:

“Mice were anesthetized using combination of ketamine (150 mg/kg) and xylazine (10 mg/kg).”

Q4. Results: photomicrographs in Figures 2 and 3 are too dark. Please, make them brighter.

A4. We appreciate the reviewer’s suggestion and recognize the importance of improving the photomicrographs in Figures 2 and 3. We have adjusted the brightness in these figures to enhance visibility. Additionally, we have corrected an error in Figure 3 by replacing panels B and C, which were previously misidentified. The updated figures now offer improved visual clarity (see revised Figures 2 and 3).

Q5. Figures 4,5 and 6: the fonts are too small. Please increase them.

A5. We have now revised Figures 4, 5, and 6 with increased font size to enhance readability and clarity. The updated figures provide improved visual clarity (see new Figure 4,5 and 6).

Q6. The conclusion needs its subheading.

A6. The manuscript has been revised accordingly, and it has been added to manuscript and is now designated as section 5.

Reviewer 2 Report

Comments and Suggestions for Authors

Review: “Dysregulation of miR-223, miR-146a, and miR-193a expression”

This is a very interesting study. The results are indicative of what the authors state. It is exciting to see such promising results and hopeful this report will encourage more trials in animal models for acute and chronic observations in the relationships of micro RNAs as biomarkers to various disease states.

A few edits the authors might consider

A few odd expressions:

1.       Line 35: “Multiple sclerosis (MS) is the conducting reason” maybe Multiple sclerosis (MS) is a primary reason….

2.       Figure 5 “miR-193a expression pattern in the brain decreased in the chronic phase of EAE” but in Part C the histogram is the highest in “chronic”. Maybe I am misreading the measure but it seems opposite of the explanation.

3.       Line 124: I would change the wording of the score explanation of 0, the current phrasing is confusing.

“No demyelination thing;”

4.       Line 318, 354, 387: Minor typo corrections recommended. 

“MiR-223 serves as a myeloid cell-specific MicroRNA          …..”

“Furth more, it”  …..Fourth more

Reword “miR-146a might be decline to precisely control its objects,”

Comments on the Quality of English Language

see above

Author Response

A few edits the authors might consider

A few odd expressions:

Q1. 1.       Line 35: “Multiple sclerosis (MS) is the conducting reason” maybe Multiple sclerosis (MS) is a primary reason….

A1. We thank the reviewer for this comment, we have updated the introduction, and this expression is no longer used.

Q2. 2.       Figure 5 “miR-193a expression pattern in the brain decreased in the chronic phase of EAE” but in Part C the histogram is the highest in “chronic”. Maybe I am misreading the measure but it seems opposite of the explanation.

A2. It was a typo error, and the legend is corrected and highlighted (“decreased” changed to “increase”).

Q3. 3.       Line 124: I would change the wording of the score explanation of 0, the current phrasing is confusing.

“No demyelination thing;”

A3. We have corrected and updated the manuscript (lines 124-126).

Q4. 4.       Line 318, 354, 387: Minor typo corrections recommended. 

“MiR-223 serves as a myeloid cell-specific MicroRNA…..”

“Furth more, it” …..Fourth more

Reword “miR-146a might be decline to precisely control its objects,”

A4. We have now revised the manuscript and corrected the typos at lines 318, 354, and 387. The text has been revised accordingly and showed with blue font.

Round 2

Reviewer 1 Report

Comments and Suggestions for Authors

all queries addressed